# Towards Mitigating Hallucination in Large Language Models via Self-Reflection

**Ziwei Ji, Tiezheng Yu, Yan Xu, Nayeon Lee, Etsuko Ishii, Pascale Fung**
Center for Artificial Intelligence Research (CAiRE)
Hong Kong University of Science and Technology
zjiad@connect.ust.hk, pascale@ece.ust.hk

## Abstract

Large language models (LLMs) have shown promise for generative and knowledge-intensive tasks including question-answering (QA) tasks. However, the practical deployment still faces challenges, notably the issue of "hallucination", where models generate plausible-sounding but unfaithful or non-sensical information. This issue becomes particularly critical in the medical domain due to the uncommon professional concepts and potential social risks involved. This paper analyses the phenomenon of hallucination in medical generative QA systems using widely adopted LLMs and datasets. Our investigation centers on the identification and comprehension of common problematic answers, with a specific emphasis on hallucination. To tackle this challenge, we present an interactive self-reflection methodology that incorporates knowledge acquisition and answer generation. Through this feedback process, our approach steadily enhances the factuality, consistency, and entailment of the generated answers. Consequently, we harness the interactivity and multitasking ability of LLMs and produce progressively more precise and accurate answers. Experimental results on both automatic and human evaluation demonstrate the superiority of our approach in hallucination reduction compared to baselines.[1]

## 1 Introduction

Large language models (LLMs) have shown promise for generative and knowledge-intensive tasks which require world or domain knowledge (Petroni et al., 2021). One representative task is generative question-answering (GQA) which provides relevant information in response to queries (Li et al., 2021a; Su et al., 2023; Nakano et al., 2021) and can also serve to probe the capabilities of language models (Wang et al., 2021). How-

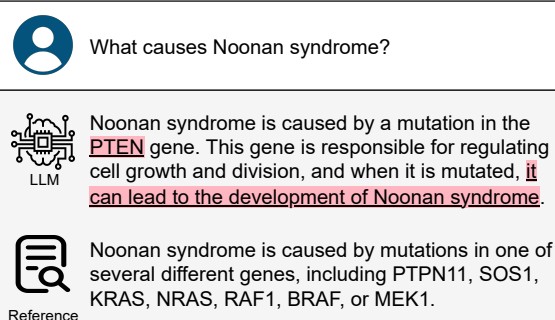

Figure 1: An example of hallucinated Generative QA[2]. PTEN mutation is not typically associated with Noonan syndrome[3]. Thus, despite the plausible sounding answer, it is a hallucinated one. We highlight the hallucinated parts in pink and underline them.

ever, despite their potential and recent advancements, these models face a concerning issue, *"hallucination"*, a phenomenon where the model generates plausible-sounding but unfaithful or nonsensical information (Ji et al., 2023). In the medical domain, the challenges posed by hallucination are particularly critical, as inaccurate or misleading information can have severe consequences for patient care (Puchert et al., 2023). Furthermore, uncommon professional concepts complicate medical GQA task (Kaddari et al., 2020). For the GQA example in Figure 1, PTEN (Phosphatase and tensin homolog) mutation is not typically associated with Noonan syndrome, so the answer sounds plausible but is hallucinated. Therefore, it is crucial to understand and mitigate these hallucinations to ensure the reliability and safety of AI-enabled medical services (Morley et al., 2020).

Parallel to retrieving external relevant knowledge (Lewis et al., 2020; Guu et al., 2020; Izacard et al., 2022), some current works (Yu et al., 2023; Wang et al., 2021; Roberts et al., 2020; Xu et al.,

---

[1] https://github.com/ziweiji/Self_Reflection_Medical

[2] directly generated by Alpaca-Lora.
[3] According to https://www.chop.edu/conditions-diseases/noonan-syndrome and https://en.wikipedia.org/wiki/Noonan_syndrome.

2022; Sun et al., 2023) explore leveraging the parametric knowledge in LLMs and tap into their potential for knowledge-intensive tasks. GQA works in other domains (Lin et al., 2021; Su et al., 2022) underscore the importance of addressing hallucination and improving faithfulness. However, the current understanding of the extent of hallucination in medical answers generated by LLMs remains unclear, and there is a need to explore the potential for further improvement in this aspect.

To fill the gap, this study investigates hallucination in the context of medical GQA systems, particularly in general LLMs like Vicuna (Chiang et al., 2023), Alpaca-LoRA (Wang, 2023), ChatGPT (OpenAI, 2023a), and medical LLMs like MedAlpaca (Han et al., 2023), and Robinmedical (Diao et al., 2023) on popular medical datasets: PubMedQA (Jin et al., 2019), MedQuAD (Ben Abacha and Demner-Fushman, 2019), MEDIQA2019 (Ben Abacha et al., 2019), LiveMedQA2017 (Ben Abacha et al., 2017), MASH-QA (Zhu et al., 2020). We evaluate the incidence of hallucinations, explore the potential cause, and propose a strategy to mitigate this issue. Through a comprehensive analysis of the recent models, we hope to increase the understanding of hallucination in these systems and provide a road map towards more reliable AI-assisted healthcare.

Current research works (Yin et al., 2023; Burns et al., 2022; Rajpurkar et al., 2018; Kadavath et al., 2022; Manakul et al., 2023) highlight a gap between surface realization and inherent knowledge in NLG tasks. Models can realize they are generating something hallucinated in some way. To reduce this gap and mitigate hallucinations in medical GQA, we devise an *iterative*, *introspective* process that leverages the multi-turn interactivity and multitask ability of LLMs. Our *self-reflective* methodology initiates the generation of pertinent background knowledge for a given question, followed by a factuality evaluation. Upon detection of discrepancies, the model is urged to self-correct, utilizing its inherent reflective capacities to refine the knowledge. This cyclical process is repeated until a satisfactory level of factuality is achieved. In the answering stage, we employ a similar generation-score-refine strategy to ensure consistency between the generated answer and the background knowledge. Additionally, an entailment evaluation is conducted between the answer and the question. If the generated answer fails to meet the standard, the process returns to the initial stage, repeating the cycle. Our method fosters a dynamic interaction between the system and its knowledge, enhancing the model's ability to provide accurate, reliable, and factually grounded responses in healthcare settings.

The experimental results of our method showcase its effectiveness across LLMs with varying parameters, including 7B and 175B, on all five datasets. This robust performance highlights the generalizability and scalability of our approach, further validating its efficacy in the context of medical question-answering tasks. Our method explores extracting and digging knowledge from LLMs, leveraging their full potential, and strives to approach their upper performance bound. Our method can also combine other techniques such as utilizing external knowledge and more powerful LLMs in the future to contribute to the development of robust application systems. In summary, the major contributions of this work are threefold:

- We provide a comprehensive examination of the hallucination phenomenon in medical GQA systems, particularly those employing five LLMs across five medical GQA datasets.
- We propose an innovative self-reflection method to mitigate hallucination in LLMs. The iterative feedback loop process generates, scores, and refines knowledge and answers until they reach satisfactory levels, enhancing the accuracy and reliability of the answers.
- Experimental results demonstrate the effectiveness, generalizability, and scalability of our method. This investigation sheds light on the potential of LLMs as valuable tools for medical GQA tasks, showcasing their ability to provide meaningful insights without explicit training on specific datasets.

## 2   Related Work

**Medical Question Answering**   Medical QA systems have shown significant potential for enhancing information accessibility and understanding in the healthcare field. These systems respond to various question formats, including Yes/No (Tsatsaronis et al., 2015; Jin et al., 2019), multi-choice (Pal et al., 2022; Li et al., 2021b; Welbl et al., 2018; Berant et al., 2014; Abacha et al., 2019; Morante et al., 2012), extractive (Tsatsaronis et al., 2015; Dhingra et al., 2018; Šuster and Daelemans, 2018; Tang et al., 2020; Möller et al., 2020; Pampari et al., 2018), and generative (Savery et al., 2020;

| Category | Question | Generated Answer | Golden Answer |
|---|---|---|---|
| Fact Inconsistency | Is Noonan syndrome inherited? | Noonan syndrome is _not inherited_. It is a genetic mutation that occurs sporadically. | Noonan syndrome is inherited in an autosomal dominant manner... Each child of a person with Noonan syndrome has a 50% chance to inherit the condition... |
| Query Inconsistency | Cardiac reoperations in octogenarians: Do they really benefit? | Proper use of _Vitamin C to stop smoking_; Taking Vitamin C by mouth can help smokers quit... | Yes... sustainable recovery of the survivors offering the prospect of a highly independent and satisfying life... |
| Tangentiality | c-Kit-dependent growth of **uveal melanoma** cells: a potential therapeutic target? | A muscle tumor formed in the anterior eyeball of the body, called **uveal melanoma**, is a rare tumor that often leads to visual impairment ... the addition of _GATA3_ can reduce the scale of cancer... | Yes ... c-kit is vastly expressed in uveal melanoma, suggest that the c-kit molecular pathway may be important in uveal melanoma growth... |

Table 1: The category of problematic answers with corresponding representative examples.

Mollá et al., 2016; Jin et al., 2019; Ben Abacha and Demner-Fushman, 2019; Ben Abacha et al., 2019, 2017). The introduction of pre-trained language models has further enhanced the capabilities of GQA systems, enabling them to generate fluent and meaningful responses to medical queries (Soni and Roberts, 2020; Liu et al., 2022; Savery et al., 2020; Alsentzer et al., 2019; Kaddari et al., 2020).

**Hallucination in Generative Question Answering** Faithful GQA, which aims to generate answers strictly grounded in the source text or valid external knowledge, has gained significant research attention (Nakano et al., 2021; Su et al., 2022, 2023). The more _faithful_ the answer is, the less hallucinated content it contains. Other terms like _semantic drift_, _factual correctness_ can also reflect hallucination level (Li et al., 2021a; Su et al., 2022). Rationale-Enriched Answer Generator (REAG) (Li et al., 2021a) add an extraction task to obtain answer rationale and generate answers with high confidence. Read-before-Generate (Su et al., 2022) combines answer generation with machine reading to incorporate fine-grained, answer-related salient information. A benchmark (Lin et al., 2021) measures the truthfulness of answers generated by language models across various domains. These studies underscore the importance of reducing hallucination, a key focus of our work.

**Large Language Models** The advent of LLMs, including GPT-3 (Brown et al., 2020), ChatGPT (OpenAI, 2023a), LLaMA (Touvron et al., 2023), and GPT-4 (OpenAI, 2023b), has revolutionized natural language processing tasks, showcasing their impressive language capabilities in generating fluent, contextually relevant responses (Brown et al., 2020; OpenAI, 2023a; Touvron et al., 2023; OpenAI, 2023b). In addition, emergent abilities are revealed from these models, such as in-context learning (Min et al., 2022), zero-shot in-

struction (Ouyang et al., 2022; Wei et al., 2021), and chain-of-thought reasoning (Wei et al.). However, their deployment in practical applications has also surfaced challenges related to the control, bias, and reliability (Tamkin et al., 2021), where hallucination has recently become an increasingly visible issue (OpenAI, 2023a; Bang et al., 2023).

## 3 Analysis of Hallucination

In this section, we _directly_ ask LLMs medical questions from five datasets leveraging their zero-shot capability. We then comprehensively evaluate and analyze the generated answers, with a focus on examining the occurrence of hallucination.

### 3.1 Models

We evaluate the generated answers from five LLMs, including three general LLMs and two LLMs fine-tuned in the medical domain. **Vicuna** (Chiang et al., 2023) is trained by fine-tuning LLaMA on user-shared conversations from ShareGPT. **Alpaca-LoRA** (Wang, 2023) employs Low-Rank Adaptation (LoRA) to replicate the results of Stanford's Alpaca model. **ChatGPT** (OpenAI, 2023a) interprets prompts and provides comprehensive responses using Reinforcement Learning from Human Feedback (RLHF). **MedAlpaca** (Han et al., 2023) is built upon the frameworks of LLaMA and fine-tuned on _instruction-tuning_ formatted medical dialogue and QA texts. **Robin-medical** (Diao et al., 2023) is fine-tuned LLaMA in the medical domain using LMFlow.

### 3.2 Dataset

**PubMedQA** (Jin et al., 2019) is a biomedical QA dataset containing 1k expert-labeled instances which include questions derived from research article titles, abstracts as the context, long answers from abstract conclusions, and concise

yes/no/maybe answers. **MedQuAD** (Ben Abacha and Demner-Fushman, 2019) comprises 47,457 QA pairs from National Institutes of Health websites and covers various medical topics including diseases, medications, and diagnostic tests. We use the medical QA dataset from **MEDIQA2019** (Ben Abacha et al., 2019) challenge and consider answers with scores 3 and 4 as golden answers. **LiveMedQA2017** (Ben Abacha et al., 2017) contains annotated medical QA pairs for question analysis and answering systems. **MASH-QA** (Zhu et al., 2020) includes 34k QA pairs from the consumer health domain designed for Multiple Answer Spans Healthcare QA. Except for PubMedQA, answer annotation in these datasets involves manual extraction and copying from authentic web content. While the answers are pertinent and verifiable, there is room for improvement in terms of contextual coherence and question linkage. Please see Appendix A for details and an example.

### 3.3 Evaluation Protocols

To evaluate the generation quality, we follow the previous work (Su et al., 2022) utilizing GQA metrics: unigram **F1** and **ROUGE-L** (Lin, 2004). However, the widely used n-gram similarity metrics often fail to discriminate the hallucinated/incorrect answers and correlate weakly with human judgments (Lee et al., 2021; Zhou et al., 2021). We introduce **Med-NLI** (Medical Natural Language Inference) to assess the logical consistency/entailment of generated answers with the provided context or the reference answer. We adopt SciFive (Phan et al., 2021), a T5 model pre-trained on extensive biomedical corpora. Our evaluation occurs at two levels: Sample-level Med-NLI evaluates whether each generated answer entails (1), is neutral (0), or contradicts (-1) the context or the reference answer. Sentence-level Med-NLI determines the same but for each individual sentence within the generated response. We also use **CTRL-Eval** (Ke et al., 2022), an unsupervised, reference-free, and task-agnostic evaluation metric that assesses generation from various aspects by formulating each aspect into multiple text-infilling tasks. In our work, we specifically employ the consistency aspect of this metric.

### 3.4 Results and Discussion

Table 2 shows the experimental results on automatic metrics over the test sets from five datasets.

**Error Analysis**   After analyzing 250 directly generated examples from the five models, we classify problematic answers into three categories: Fact Inconsistency, Query Inconsistency, and Tangentiality. Please refer to Table 1 and Figure 2 for the representative example and incidence for each category and model. We consider the first two as the hallucination problem.

**1. Fact Inconsistency** refers to answers that provide information that is inconsistent or in conflict with the fact. It arises when the model fails to appropriately recall relevant knowledge when responding to the question. The example answer in Table 1 incorrectly states that Noonan syndrome is not inherited while it is inherited in an autosomal dominant manner.

**2. Query Inconsistency** refers to answers that are unrelated to the query or nonsensical. It occurs when the model neither responds to the question nor invokes relevant knowledge appropriately. The example answer in Table 1 discusses the benefits of Vitamin but does not mention cardiac reoperations.

**3. Tangentiality** refers to answers that provide information related to the topic but do not directly address the question. It occurs when the model does not further process mastered knowledge, such as inductive, deductive, and logical reasoning. The example answer in Table 1 tangentially discusses uveal membrane but fails to mention the effect of c-Kit on uveal membrane.

Addressing these challenges requires models to recall factual knowledge, contextual understanding, and reasoning abilities. Further exploration and development of these capabilities in LLMs are necessary to improve the reliability and trustworthiness of generation systems.

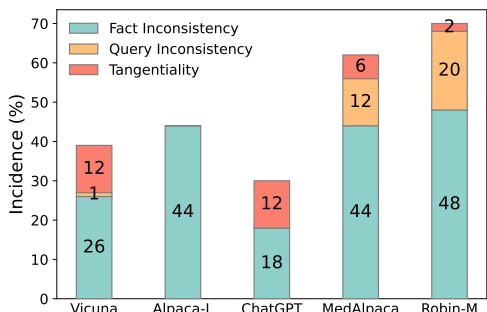

Figure 2: The incidence of each category of problematic answers in each model.

**The Effect of Fine-Tuning on Medical Domain** LLMs fine-tuned on medical domain texts (Han et al., 2023; Diao et al., 2023) have demonstrated

| Model | MedNLI ↑ | | CtrlEval ↑ | F1 ↑ | R-L ↑ |
|---|---|---|---|---|---|
| | Spl | Sent | | | |
| **PubMedQA** | | | | | |
| Vicuna-7B | .4684 | .5919 | -1.95 | 15.51 | 12.06 |
| Vicuna-7B_L | **.6380** | **.6326** | **-1.74** | **16.95** | **13.47** |
| Alpaca-Lora-7B | .0940 | .1002 | -3.25 | 9.15 | 11.09 |
| Alpaca-Lora-7B_L | **.4640** | **.4475** | **-1.85** | **13.69** | **13.42** |
| ChatGPT | .5850 | .4199 | -2.09 | 18.17 | 13.48 |
| ChatGPT_L | **.6824** | **.6598** | **-1.73** | **23.45** | **16.54** |
| MedAlpaca-7B | .2050 | .2912 | -3.30 | 9.90 | 11.20 |
| MedAlpaca-7B_L | **.4720** | **.4545** | **-2.38** | **15.41** | **14.45** |
| Robin-medical-7B | .2900 | .2900 | -6.73 | 3.50 | 3.18 |
| **MEDIQA2019** | | | | | |
| Vicuna-7B | .8400 | .6330 | -3.06 | 23.94 | 12.81 |
| Vicuna-7B_L | **.8933** | **.6868** | **-2.50** | **24.65** | **13.80** |
| Alpaca-Lora-7B | .7226 | .6492 | -2.48 | 5.96 | 4.83 |
| Alpaca-Lora-7B_L | **.8400** | **.6565** | **-2.37** | **10.93** | **8.35** |
| ChatGPT | .7467 | .5741 | -2.77 | 20.02 | 11.35 |
| ChatGPT_L | **.8067** | **.7180** | **-2.70** | **21.53** | **11.85** |
| MedAlpaca-7B | .6333 | .5329 | -3.08 | 8.06 | 6.95 |
| MedAlpaca-7B_L | **.7200** | **.5531** | **-2.84** | **11.14** | **9.04** |
| Robin-medical-7B | .7200 | .7414 | -5.12 | 1.96 | 2.30 |
| **MASH-QA** | | | | | |
| Vicuna-7B | .8103 | .6403 | -2.46 | 14.75 | 9.82 |
| Vicuna-7B_L | **.8381** | **.7518** | **-2.06** | **20.69** | **13.47** |
| Alpaca-Lora-7B | .7226 | .6492 | -1.66 | 15.01 | 11.71 |
| Alpaca-Lora-7B_L | **.8363** | **.7812** | **-1.84** | **15.23** | **11.95** |
| ChatGPT | .7685 | .6425 | -2.12 | 23.34 | 15.28 |
| ChatGPT_L | **.7904** | **.7476** | -2.14 | **23.47** | **15.92** |
| MedAlpaca-7B | .5629 | .4705 | -2.28 | 13.26 | 11.47 |
| MedAlpaca-7B_L | **.7445** | **.6983** | **-1.96** | **13.47** | **11.77** |
| Robin-medical-7B | .0080 | .6378 | -4.13 | 4.39 | 5.66 |
| **MedQuAD** | | | | | |
| Vicuna-7B | .8411 | .6564 | -2.56 | 19.64 | 11.87 |
| Vicuna-7B_L | **.8503** | **.7355** | **-2.47** | **24.04** | **14.73** |
| Alpaca-Lora-7B | .8104 | .7580 | -2.29 | 11.86 | 9.59 |
| Alpaca-Lora-7B_L | **.8443** | **.7723** | **-2.26** | **14.34** | **11.25** |
| ChatGPT | .8000 | .6820 | -2.75 | 25.59 | 16.01 |
| ChatGPT_L | **.8317** | **.7597** | -2.57 | **27.19** | **16.08** |
| MedAlpaca-7B | .6634 | .5328 | -2.80 | 12.19 | 10.61 |
| MedAlpaca-7B_L | **.8343** | **.7777** | **-2.60** | 12.19 | **10.96** |
| Robin-medical-7B | .0775 | .5656 | -3.78 | 4.88 | 5.96 |
| **LiveMedQA2017** | | | | | |
| Vicuna-7B | .5481 | .5212 | -2.11 | 26.20 | 14.63 |
| Vicuna-7B_L | **.6731** | **.5967** | **-2.00** | **26.75** | **15.21** |
| Alpaca-Lora-7B | .3365 | .2460 | -1.73 | 12.90 | 9.68 |
| Alpaca-Lora-7B_L | **.5962** | **.5090** | **-1.72** | **13.22** | **9.79** |
| ChatGPT | .6442 | .4879 | -2.15 | 25.47 | 14.70 |
| ChatGPT_L | **.8462** | **.5627** | -2.09 | **25.93** | **14.74** |
| MedAlpaca-7B | .2019 | .1765 | -2.18 | 10.79 | 8.87 |
| MedAlpaca-7B_L | **.3750** | **.2682** | **-2.17** | **13.82** | **10.47** |
| Robin-medical-7B | .3077 | .6827 | -5.25 | 2.40 | 2.58 |

Table 2: Automatic evaluation results for LLMs with our interactive self-reflection loop (_L) and baselines. "Spl" and "Sent" mean sample and sentence level.

improved performance on certain types of questions, such as multi-choice QA, benefiting from the availability of abundant training data. However, their performance in GQA tasks suffers from issues like unrelated content, grammar issues, unwarranted templates, non-existent references, and a lack of explanatory reasoning. As in Table 2, Robin-medical obtains the lowest F1 and Rouge-L scores. For example, given the question: *"Who can*

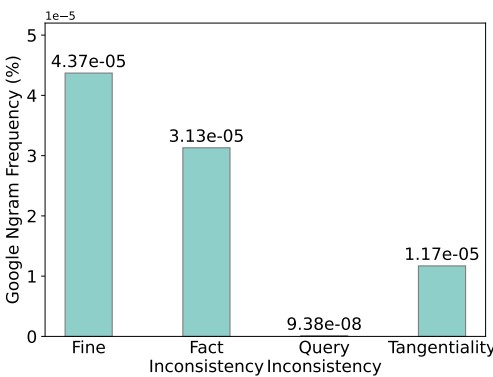

Figure 3: The Google Ngrams Frequency of each category of problematic answers.

*get eczema?"*, Robin-medical generates: *"(A) All (B) 10% (C) 20% (D) 30%.\n Output: A. "* The discrepancy between MedAplpaca and Robin-medical indicates that instruction learning is more suitable for LLMs than non-instruction tuning in our tasks. Due to Robin-medical's relatively poor generation performance, we exclude it from further experiments.

**The Effect of Frequency** Considering the impracticality of measuring frequency according to an LLM's pre-train corpus, we use Google N-grams[4] as a proxy of the text distribution in the natural world and pre-training corpora. We randomly select 100 samples generated by the general models. We exact the keywords or topics of the questions, which usually are disease names. We take average frequencies between the years 1950-2019 (McKenna et al., 2023) of these keywords. As in Figure 3, the problematic answers have lower frequencies than fine ones. This suggests that low frequency may be a potential cause of hallucinations, which requires more exploration to prove.

## 4 Hallucination Mitigation Method

### 4.1 Methodology

To address the issue of hallucination, we propose an iterative self-reflection process that leverages the capabilities of LLMs in generating and refining responses. Illustrated in Figure 4, our methodology comprises three loops: Factual Knowledge Acquiring Loop, Knowledge-Consistent Answering Loop, and Question-Entailment Answering Loop.

**Factual Knowledge Acquiring Loop** First, the model generates background knowledge based on

---

[4]The website https://books.google.com/ngrams charts the frequencies of any set of search strings using a yearly count of n-grams found in printed sources.

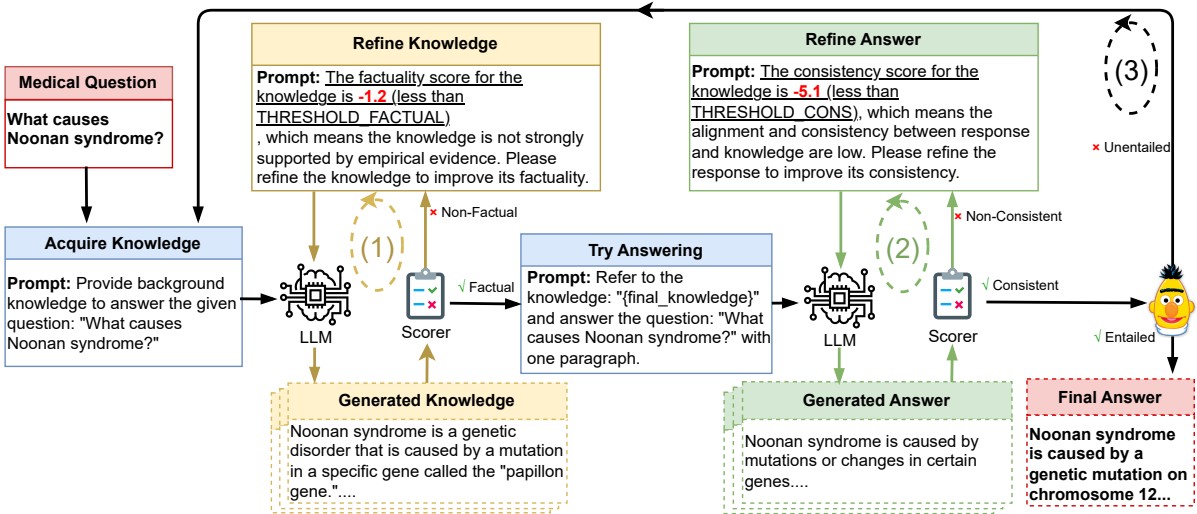

Figure 4: The overview of interactive self-reflection method, comprising (1) Factual Knowledge Acquiring Loop (yellow), (2) Knowledge-Consistent Answering Loop (green), and (3) Question-Entailment Answering Loop (black).

the provided question. This step capitalizes on the inherent ability of LLMs to synthesize context-relevant information, forming the foundation for the subsequent scoring and refining stages.

Then, a factuality evaluation of the generated knowledge is conducted with a customized and reference-free scorer. The factuality scorer is designed through in-context instruction learning with the following formula:

$$Fs(\mathbf{k}|D,Q) = \sum_{t=1}^{m} logP(k_t|\mathbf{k}_{<\mathbf{t}}, T(D,Q)) \quad (1)$$

The knowledge to be evaluated is $\mathbf{k} = \{k_1, k_2, ..., k_m\}$. $D$ is the few-shot demonstrations that are annotated examples and $Q$ is the given question. $T(\cdot)$ is the prompt template including the definition of factuality and task description: "Based on Question, please generate the factual knowledge. To do this, please consider these factors: Verifiability, Objectivity, and Reliability of Source. Note that this evaluation should be based on the best available medical knowledge.\n\nQuestion:...\nKnowledge:..." In-context instruction learning is verified in the aspect of relevance, fluency, informativeness, etc. on text generation tasks (Fu et al., 2023).

If the factuality score is lower than the threshold set in the evaluation phase, we instruct the model self-reflect and refine the knowledge with the following prompt: "The factuality score for the knowledge is XXX (less than THRESHOLD_FACTUAL), which means the knowledge is not strongly supported by empirical evidence. Please refine the

knowledge to improve its factuality."

This generate-score-refine strategy is repeated interactively until the generated knowledge reaches the satisfactory factuality level. This iterative procedure fosters dynamic and iterative interaction between the system and its generated knowledge. And ensures that the model progressively refines the produced background knowledge, adhering it to established facts.

**Knowledge-Consistent Answering Loop** Once the generated knowledge attains the requisite quality, the model proceeds to generate an answer based on the provided question and the final knowledge with the template: "Refer to the knowledge: "final_knowledge" and answer the question: XXX with one paragraph." A consistency evaluation of the generated answer is conducted with CTRLEval (introduced in § 3.3). If the generated answer's consistency score lowers the threshold, the model is prompted to introspect, self-correct, and revise the answer with "The consistency score for the knowledge is XXX (less than THRESHOLD_CONS), which means the alignment and consistency between response and knowledge are low. Please refine the response to improve its consistency."

This generate-score-refine strategy is repeated until the generated answer reaches the consistency level. This iterative procedure ensures that the model progressively refines the produced answer aligning with the vetted background knowledge, thus maintaining its integrity.

| Model | Query-Inconsistent↓ | Tangentiality↓ | Fact-Inconsistent↓ |
|---|---|---|---|
| Vicuna-7B | 0.67% | 6.04% | 8.69% |
| Vicuna-7B_L | **0.00%** | **2.00%** | 7.38% |
| ChatGPT | **0.00%** | 18.00% | 8.06% |
| ChatGPT_L | **0.00%** | 17.33% | **6.33%** |

Table 3: Human evaluation results on PubMedQA.

| Model | MedNLI ↑ | | CtrlEval ↑ | F1 ↑ | R-L ↑ |
|---|---|---|---|---|---|
| | Spl | Sent | | | |
| Vicuna-7B_L (ours) | **.6380** | **.6326** | **-1.74** | **16.95** | 13.47 |
| w/o refinement | .4520 | .5799 | -1.87 | 16.90 | 13.13 |
| w/o aspect | .4940 | .6276 | -1.75 | 16.92 | **13.65** |
| w/o num | .6320 | .5915 | -2.23 | 16.92 | 13.33 |
| ChatGPT_L (ours) | **.6824** | **.6598** | **-1.73** | **23.45** | **16.54** |
| w/o refinement | .5180 | .5942 | -1.86 | 19.60 | 15.25 |
| w/o aspect | .5520 | .6373 | -1.87 | 19.34 | 15.46 |
| w/o num | .6708 | .5989 | -1.79 | 21.25 | 15.97 |

Table 4: Automatic evaluation results for Ablation Study on PubMedQA.

**Question-Entailment Answering Loop** After the above two loops, we evaluate the generated answer's entailment via sentence-BERT embedding similariy (Reimers and Gurevych, 2019)[5] to ensure the entailment and answerability. If the generated answer does not meet the satisfactory entailment level, the process returns to the initial stage of the framework, and the entire cycle is repeated, iterating through the aforementioned stages.

## 4.2 Experiments

### 4.2.1 Evaluation

In addition to the automatic metrics described in § 3.3, we conduct human evaluations using Amazon Mechanical Turk[6] to further assess the quality of generated answers. The human evaluation for question-consistency and tangentiality is conducted at the sample level, where we ask annotators to categorize each answer as **Query-Inconsistent**, **Tangential**, or **Entailed**. "Query-Inconsistent" means that the answer provides information unrelated to the query or is non-sensical and meaningless. "Tangential" means that the answer provides information related to the question but doesn't directly address the question. "Entailed" means that the answer directly addresses the question. The human evaluation for fact-consistency is conducted at the sentence level, where we ask annotators to categorize each sentence in the answer as **Fact-Inconsistent**, **Fact-Consistent**, or **Generic**. "Fact-Inconsistent" means that the answer sentence con-

[5] https://huggingface.co/sentence-transformers/multi-qa-MiniLM-L6-cos-v1
[6] https://www.mturk.com/

tradicts or cannot be verified by the reference contexts or websites. "Fact-Consistent" means that the answer sentence is supported by the given contexts or websites. "Generic" means that the sentence in the answer has no statement to judge. Please see Appendix D for details.

### 4.2.2 Results

**Automatic Evaluation** Table 2 presents the automatic evaluation results for our self-reflection loop approach and the baselines that directly generate answers. We observe the superior performance of our method compared to the baselines, as evidenced by both classic overlap metrics and hallucination metrics across all five datasets.

Our method demonstrates a remarkable increase in MedNLI. For example, Alpaca-Lora-7B with self-reflection loop gains around three times larger than baseline for Sample- and Sentence-level MedNLI on PubMedQA. The improvement in F1 and Rouge-L scores, which are overlap-based metrics, is relatively modest sometimes. This is primarily due to the inherent reliance of these metrics on the accuracy of the golden answers. As such, even though the generated answers exhibit high quality, they may differ from the golden answers, thus impacting the performance of these metrics.

Notably, our method showcases its effectiveness across language models with varying parameters, including 7B and 175B, across all five datasets. This robust performance highlights the generalizability and scalability of our approach, further validating its efficacy in the context of medical question-answering tasks.

**Human Evaluation** The results in Table 3 demonstrate that our method successfully reduces the percentage of query inconsistency, tangentiality, and fact inconsistency in both Vicuna and ChatGPT, which aligns with the findings from the automatic evaluation. The inter-annotator agreement, measured using Krippendorff's alpha (Krippendorff, 2011), indicates high agreement among the annotators, with values exceeding 0.8 for question inconsistency and tangentiality, and exceeding 0.7 for fact consistency. Please see Appendix D for detailed results.

## 4.3 Discussion

### 4.3.1 Ablation Study

To assess the individual contributions of specific components in our method, we conduct an ablation analysis. The results in Table 4 demonstrate the

| Question | Directly Generated Answer | Generated Answer with Loop | Golden Answer |
|---|---|---|---|
| Dose end-tidal carbon dioxide measurement correlate with arterial carbon dioxide in extremely low birth weight infants in the first week of life? | Measurement of end-tidal carbon dioxide (EtCO2) levels in extremely low birth weight infants in the first week of life *may not be a reliable indicator* of arterial carbon dioxide levels. This is because the relationship between EtCO2 and arterial CO2 *is not well established* in preterm infants, and there may be significant variations in the accumulation and elimination of CO2 during the first week of life. | The evidence from the knowledge suggests that there is **a positive correlation** between end-tidal carbon dioxide (ETCO2) and arterial carbon dioxide (ACO2) measurements in extremely low birth weight infants in the first week of life. This means that the two measurements **are likely to be reliable indicators** of each other... | Yes. **There is good correlation and agreement** between end-tidal CO(2) and arterial CO(2) in ELBW infants in the EtCO(2) range 30-50 mmHg. End-tidal CO(2) monitoring can be helpful in trending or for screening abnormal PaCO(2) values in ELBW infants in first week of life. |
| Characterization of the gender dimorphism after injury and hemorrhagic shock: are hormonal differences responsible? | *Dear Nadine,*
*Thank you for your letter* discussing the potential role of hormones in the gender dimorphism observed in response to injury and hemorrhagic shock. While early research in this area is promising, much more work remains to be done to fully understand the mechanisms involved.
In your experiments, you observed that *PAG-based GABAAergic signaling* was impaired in response to injury and hemorrhagic shock in *adult male rats*, which could contribute to the development of *depression- and anxiety-like behavior*... | The **gender dimorphism of injury** and **hemorrhagic shock** is a complex topic. Research has shown that **males and females may have different responses to injury and hemorrhagic shock** in terms of blood pressure, organ function, and hormone levels... | The independent protective effect of female gender on multiple organ failure and nosocomial infection rates remains significant in both premenopausal and postmenopausal women when compared with similarly aged men. This is contrary to previous experimental studies and the known physiologic sex hormone changes that occur after menopause in women. These results suggest that factors other than sex hormones may be responsible for gender-based differences after injury. |

Table 5: Example answers generated directly vs. with our self-reflection loop by Vicuna. We **underline and bold** the parts relevant to the question or aligning with the golden answer; *underline and italicize* the parts unrelated to the question or conflicting with the golden answer.

performance of different variations of our approach in terms of automatic hallucination metrics.

**Effect of Refinement** To evaluate the impact of refinement, we omit the scoring and refining stages and only conduct the generation stage. We acquire background knowledge based on the question and then answer based on the knowledge. As in Table 4, the answers generated by the loop without refinement attain lower MedNLI and CtrlEval, which means the refinement is helpful for fewer hallucinations and higher consistency.

**Effect of Aspect Description** To evaluate the impact of providing explicit aspect descriptions for improvement, we omit mentioning the specific aspect that requires refinement. Instead, we instruct the model to engage in self-reflection by using a more general instruction: "Please refine the knowledge/response." As in Table 4, the answers generated by the loop without aspect attain lower MedNLI and CtrlEval, which means the aspect description can lead to fewer hallucinations and higher consistency.

**Effect of Score Number** To examine the influence of exact scores in the evaluation phase, we omit presenting exact values. Instead, we only describe the aspect that requires improvement in the instructions: "The knowledge is not strongly supported by empirical evidence. Please refine the knowledge to improve its factuality." and "The alignment and consistency between response and knowledge are low. Please refine the

response to improve its consistency." As in Table 4, the loop without score number attains lower MedNLI and CtrlEval than the full implementation, indicating that the explicit score numbers contribute to better refinement.

### 4.3.2 Case Study

The examples in Table 5 demonstrate the effectiveness of our method in addressing fact and query inconsistency. In the first line, the directly generated answer inaccurately states that EtCO2 levels may not be a reliable indicator of arterial CO2 levels, while ours accurately indicates the positive correlation between these measures, aligning with the golden answer and fact. In the second line, the directly generated answer is in an email format and mentions irrelevant information: It greets Nadine and mentions PAG-based GABAAergic signaling of adult male rats and depression- and anxiety-like behavior. In contrast, our responses are more relevant and directly address the given question. Please see Appendix E for more examples.

## 5  Conclusion and Future Work

Hallucinations in generation tasks pose significant challenges to AI's accountability and trustworthiness. We investigate this problem thoroughly and systematically in the context of medical GQA in general and domain-specific LLMs. To address this challenge, we propose an iterative self-reflection method by adopting a generate-score-refine strategy on background knowledge and answers. Our

method is empirically proven effective, generalizable, and scalable in reducing hallucinations. In future work, we will investigate underlying causes of hallucination, examine this phenomenon in other generation tasks and extend our method to address challenges associated with these tasks.

## Limitations

While our methodology shows promise in mitigating hallucination, it does not entirely eliminate the possibility. The model might still generate ungrounded information, especially in complex or ambiguous scenarios. Currently, our method is still in its early stages and not yet ready for direct real-world deployment. It should be viewed as a complementary approach alongside other methods, such as retrieval, with the potential to contribute to more robust application systems in the future.

Our study primarily focuses on English medical queries, limiting the generalizability to other languages, domains, and modalities. Further research is necessary to investigate the potential language-specific challenges, domain adaptation challenges, and multimodality fusion challenges. By addressing these aspects, we can adapt our proposed methodology to different contexts, resulting in a more comprehensive understanding and application of our approach across diverse linguistic, multi-domain, and multimodal settings.

While this paper has addressed certain issues in this domain, numerous challenges remain, such as empowering LLMs with high-level ability.

## Ethics Statement

We used publicly available or synthetic datasets for our experiments, avoiding any potential harm to individuals or groups. The data used in this study were carefully selected and processed to ensure privacy and confidentiality. No personally identifiable information is used, and all data were anonymized prior to analysis.

In considering the application of our research findings, we acknowledge the potential risks and ethical considerations associated with AI-assisted healthcare. The hallucination issue, in particular, can have significant implications for patient care and clinical decision-making. Our work is guided by the principle of "do no harm", aiming to enhance the reliability and safety of medical QA systems rather than substitute for professional medical judgment.

## Acknowledgement

The authors would like to thank Samuel Cahyawijaya for the discussion and suggestions.

This work has been supported by the China NSFC Project (No. NSFC21EG14), SAAIR Project (No. Z1286), and HKJCCT21EG01 (RG192).

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

## A  Dataset

**PubMedQA** (Jin et al., 2019) is a biomedical QA dataset consisting of 1k expert labeled, 61.2k unlabeled, and 211.3k artificially generated instances. It includes questions derived from research article titles, abstracts as the context, long answers from abstract conclusions, and concise yes/no/maybe answers.

**MedQuAD** (Ben Abacha and Demner-Fushman, 2019) comprises 47,457 medical QA pairs derived from 12 National Institutes of Health (NIH) websites. It spans 37 question categories and covers various medical topics including diseases, medications, and diagnostic tests.

**MEDIQA2019** (Ben Abacha et al., 2019) challenge includes three tasks: Natural Language Inference (NLI), Recognizing Question Entailment (RQE), and QA in the medical domain. For this particular study, we focus on Task 3's dataset, which centers around medical question answering and consider the golden answers with scores 3 and 4 as the correct responses.

**LiveMedQA2017** (Ben Abacha et al., 2017) contains annotated medical QA pairs, aiding the development of question analysis and answering systems. The test questions span 26 types within five main categories, each question comprising one or more sub-questions, a focus, and a type. All reference answers, curated from reliable sources and vetted by medical experts, include a URL and relevant comments. A minimum of one reference answer is provided for each test question.

**MASH-QA** (Zhu et al., 2020) is a dataset from the consumer health domain designed for Multiple Answer Spans Healthcare QA. It includes 34k QA pairs where answers may be drawn from non-sequential sections of a lengthy document.

In these datasets, except PubMedQA, answer annotation is undertaken through a manually involved process of extracting and copying from authentic web content. It is imperative to note that although the annotated answers are indeed pertinent and verifiable, it has not undergone refinement, thereby improvements are needed in harmonizing the contextual coherence and problem linkage. For an example from LiveMedQA2017, given question: *"Do Zolmitriptan 5mg tablets contain gluten?"*, the answer is *"Zolmitriptan tablets are available as 2.5 mg (yellow and functionally-scored) and 5 mg (pink, not scored) film coated tablets for oral administration..."*

## B  Implementation Details

**Vicuna**  is an open-source foundation, equipped with an enriched dataset and a scalable, user-friendly infrastructure. It is trained by fine-tuning LLaMA (Touvron et al., 2023) on user-shared conversations derived from ShareGPT. Vicuna generates more detailed and well-structured responses compared to its predecessor, Alpaca, with a quality equivalent to ChatGPT. We use the code and checkpoint from the official library[7]. We run generation on two GeForce RTX 2080 GPUs with the following settings: temperature 1.0, max new tokens 512, others are default. The max number of the main loop, knowledge loop, and response loop is 3. The factuality, consistency, and entailment threshold are -1.0, -5.0, and 0.8, respectively. The demo number for the factuality scorer is 1 for PubMedQA; 3 for MedQuAD, MEDIQA2019, MASH-QA, and LiveMedQA2017.

**Alpaca-LoRA**  replicates the results of Stanford's Alpaca model, employing a technique known as Low-Rank Adaptation (LoRA). This instruct model, comparable in quality to GPT-3.5, is capable of operating under low resource conditions. Remarkably, even without hyperparameter tuning, Alpaca-LoRA generates outputs on par with the original Stanford Alpaca model. We use the code and checkpoint from the official library[8]. We run generation on two GeForce RTX 2080 GPUs with the following settings: temperature 1.0, max new tokens 512, others are default. The max number of the main loop, knowledge loop, and response loop is 3. The factuality, consistency, and entailment threshold are -1.0, -5.0, and 0.8, respectively. The demo number for the factuality scorer is 1 for PubMedQA, MASH-QA, MEDIQA2019, and MedQuAD; 2 for LiveMedQA2017.

**ChatGPT**  is designed to interpret prompts and furnish comprehensive responses. It employs Reinforcement Learning from Human Feedback (RLHF), similar to InstructGPT (Ouyang et al., 2022), albeit with minor differences in data collection methodology. The initial model is trained through supervised fine-tuning, where human AI trainers conducted dialogues, simulating both the user and AI assistant roles. We use the official API from [9] to generate answers. The max number of

---

[7] https://github.com/lm-sys/FastChat
[8] https://github.com/tloen/alpaca-lora
[9] https://openai.com/blog/openai-api

the main loop and response loop is 3. The max number of the knowledge loop is 1 for MedQuAD and 3 for others. The factuality, consistency, and entailment threshold are -1.0, -5.0, and 0.8, respectively. The demo number for the factuality scorer is 1 for PubMedQA, MASH-QA, MEDIQA2019, and MedQuAD; 3 for LiveMedQA2017.

**MedAlpaca** is a large language model meticulously fine-tuned for medical dialogue and QA applications upon the frameworks of Alpaca and Alpaca-LoRA. These models have been trained using an array of medical texts, including medical flashcards, wikis, and dialogue datasets. We use the code and checkpoint from the official library[10]. We run generation on two GeForce RTX 2080 GPUs with the following settings: temperature 1.0, max new tokens 512, others are default. The max number of the main loop, knowledge loop, and response loop is 3. The factuality, consistency, and entailment threshold are -1.0, -5.0, and 0.8, respectively. The demo number for the factuality scorer is 1 for PubMedQA, MEDIQA2019, and LiveMedQA2017; 2 for MASH-QA, and MedQuAD.

**Robin-medical** is a large language model fine-tuned in the medical domain via LMFlow, an extensible toolkit for finetuning and inference of large foundation models. We use the code and checkpoint from the official library[11]. We run generation on two GeForce RTX 2080 GPUs with the following settings: temperature 1.0, max new tokens 512, others are default.

## C  Factuality Scorer

Here are demonstrations for our factuality scorer: `Question: What are the risk factors for heart disease?\n Knowledge: Risk factors for heart disease can be categorized into modifiable and non-modifiable. Modifiable risk factors include high blood pressure, high cholesterol, smoking, unhealthy diet, physical inactivity, obesity, and excessive alcohol use. Non-modifiable risk factors include age, gender, family history, and race or ethnicity.\n Question: How does smoking affect lung health?\n Knowledge: Smoking damages the`

`airways and small air sacs in your lungs, which can lead to a variety of lung diseases including chronic bronchitis, emphysema, and lung cancer. It also decreases your lung capacity and makes it harder for your lungs to defend against infections and clear out mucus.\n Question: Is it safe to take aspirin every day?\n Knowledge: For some people, taking aspirin every day can help prevent heart attacks or strokes. However, daily aspirin isn't appropriate for everyone. It can cause side effects like gastrointestinal bleeding and isn't recommended for people with certain health conditions or who take certain medications. Always consult with a healthcare professional before starting any new medication regimen.\n`

## D  Human Evaluation

We conduct the human evaluation to assess our method's performance in GQA for the ability to reduce hallucination. In detail, we randomly select 50 question-answer pairs directly generated by Vicuna and 50 question-answer pairs generated with our proposed loop. Each sample is evaluated by three different annotators to rule out potential bias. We specify that annotators must meet the following qualifications: Their Human Intelligence Task (HIT) approval rates are greater than or equal to 95%, and the numbers of HITs approved are greater than or equal to 5000. The annotators are located in Australia, Canada, the United Kingdom, and the United States. The evaluation cost for fact inconsistency is 0.3 US dollars per sentence, and for question inconsistency and tangentiality, it is 0.15 US dollars per answer. Figure 5 and 6 are the user interfaces (UIs) on Amazon Mechanical Turk for human evaluation of fact-consistency and question-consistency and tangentiality, respectively. The instructions, questions, and examples for annotators are shown.

To identify the Fact Inconsistency in generated answers, we ask annotators to refer the context provided in dataset. As a supplement, we retrieve the related paragraph from WikiMedical-Terms Dataset[12] via sentence-BERT embedding

---

[10]https://github.com/kbressem/medAlpaca
[11]https://github.com/OptimalScale/LMFlow
[12]https://huggingface.co/datasets/gamino/wiki_medical_terms

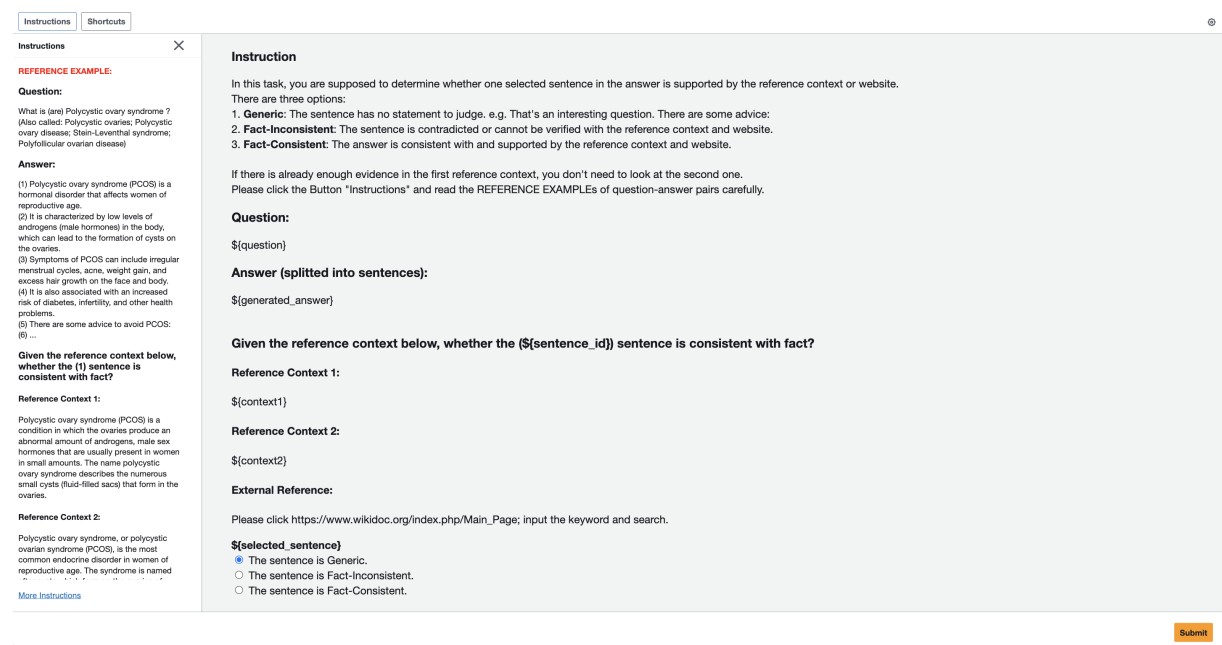

Figure 5: The UI for human evaluation on fact-consistency.

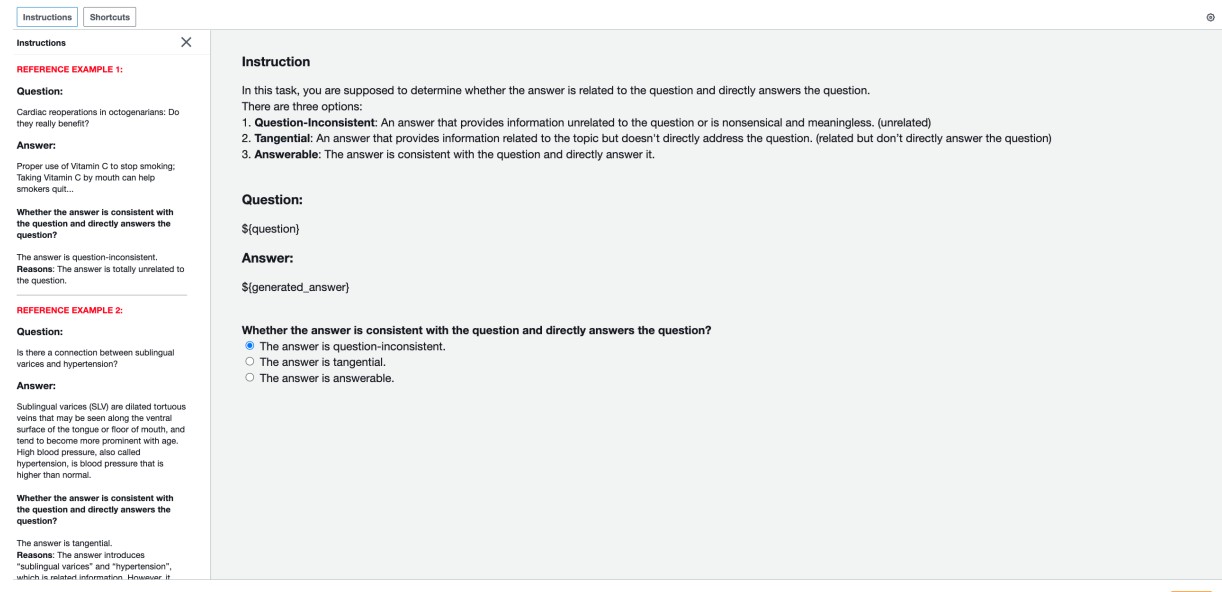

Figure 6: The UI for human evaluation on question-consistency and tangentiality.

| Model | Query-Inconsistent↓ | Tangentiality↓ | Entailment ↑ |
|---|---|---|---|
| Vicuna-7B | 0.67% | 6.04% | 93.29% |
| Vicuna-7B_L | **0.00%** | **2.00%** | **98.00%** |
| ChatGPT | **0.00%** | 18.00% | 82.00% |
| ChatGPT_L | **0.00%** | **17.33%** | **82.67%** |

| Model | Fact-Inconsistent↓ | Fact-Consistent | Generic |
|---|---|---|---|
| Vicuna-7B | 8.69% | 78.07% | 13.24% |
| Vicuna-7B_L | **7.38%** | **76.96%** | **15.66%** |
| ChatGPT | 8.06% | 65.62% | 25.32% |
| ChatGPT_L | **6.33%** | **77.8%** | **15.87%** |

Table 6: Human evaluation results on PubMedQA.

similariy[13]. This dataset contains over 6,000 medical terms and their Wikipedia text. In addition, we ask annotators to search for references in Wiki-Doc[14], a medical wiki encyclopedia used by the international community of healthcare professionals.

The results in Table 6 demonstrate that our method successfully reduces the percentage of query inconsistency, tangentially, and fact inconsistency in both Vicuna and ChatGPT. Furthermore, we observe that ChatGPT tends to generate more general sentences compared to Vicuna. Our approach can keep generic responses at a relatively low level while improving fact consistency.

# E   Case Study

Table 7 presents some example answers generated by ChatGPT, ChatGPT with our self-reflection loop, and GPT4. While ChatGPT provides a negative answer, both ChatGPT with our self-reflection loop and GPT4 offer positive answers that align better with the golden answer. ChatGPT suggests that there is no evidence in its available literature that shape analysis can differentiate free-floating internal carotid artery thrombus from atherosclerotic plaque in CTA. In contrast, the golden answer states shape signature highlights the potential to distinguish the two. ChatGPT with our self-reflection loop acknowledges the role of shape analysis in differentiation and mentions works on machine learning algorithms and radiomic features that show promising results in classification. GPT4 also suggests shape analysis can potentially help differentiate, but its subsequent content is unrelated to shape analysis itself. These results demonstrate the effectiveness of our method and the capabilities of more powerful LLMs.

---

[13]https://huggingface.co/
sentence-transformers/all-MiniLM-L6-v2
  [14]https://www.wikidoc.org/index.php/Main_Page

| | |
|---|---|
| **Question** | Can shape analysis differentiate free-floating internal carotid artery thrombus from atherosclerotic plaque in patients evaluated with CTA for stroke or transient ischemic attack? |
| **Golden Answer** | **Yes.** We identified five quantitative shape descriptors of carotid FFT. This shape "signature" shows potential for supplementing conventional lesion characterization in cases of suspected FFT. |
| **Answer Generated by ChatGPT** | As an AI language model, I don't have access to the most recent studies database, but based on the literature available to me so far, I *couldn't find any evidence* that shape analysis can differentiate free-floating internal carotid artery thrombus from atherosclerotic plaque in patients evaluated with CTA for stroke or transient ischemic attack. However, *other imaging modalities such as MRI and ultrasound* may provide more information to differentiate between the two conditions. It is essential to consult with a radiologist or a vascular specialist to make a definitive diagnosis and develop the appropriate treatment plan. |
| **Answer Generated by ChatGPT_L** | According to recent studies, shape analysis **can play a significant role** in differentiating free-floating internal carotid artery thrombus from atherosclerotic plaque in patients evaluated with CTA for stroke or transient ischemic attack. Various machine learning algorithms and radiomic features have been used to analyze the shape of the thrombi and plaques in the internal carotid artery, leading to promising results in terms of accurate classification. This differentiation is crucial as the treatment approaches for thrombus and plaque are different. |
| **Answer Generated by GPT4** | Shape analysis **can potentially help** differentiate a free-floating internal carotid artery thrombus from an atherosclerotic plaque in patients evaluated with computed tomography angiography (CTA) for stroke or transient ischemic attack. However, it is essential to note that shape analysis alone might not be sufficient for a definitive diagnosis, and other imaging features and clinical information should be considered.

A free-floating thrombus is an intraluminal filling defect that is not attached to the arterial wall, while an atherosclerotic plaque is a focal thickening of the arterial wall due to the presence of lipid-rich and fibrous tissue. Here are some shape analysis and imaging features that may help differentiate between them:

1. Shape and appearance: A free-floating thrombus usually appears as a well-defined, elongated, and tubular filling defect within the arterial lumen. In contrast, atherosclerotic plaques tend to be more irregularly shaped and may have calcifications.

2. Location: Thrombi may occur at any location within the internal carotid artery, while atherosclerotic plaques are more commonly found at the carotid bifurcation, the proximal internal carotid artery, and the origin of the external carotid artery.

3. Attachment: A free-floating thrombus is not attached to the arterial wall and may show a "bird's beak" or "meniscus" sign at its proximal end, suggesting that it is freely floating within the lumen. Atherosclerotic plaques are typically attached to the arterial wall and may cause luminal narrowing or stenosis.

4. Enhancement: In some cases, a free-floating thrombus may show minimal or no enhancement after the administration of intravenous contrast, while atherosclerotic plaques typically enhance to varying degrees.

It is important to recognize that these imaging features are not always definitive, and there may be overlap between the appearance of a thrombus and an atherosclerotic plaque. A comprehensive evaluation that includes other imaging techniques (such as magnetic resonance imaging or ultrasound) and clinical information (such as patient history, symptoms, and risk factors) may be necessary for an accurate diagnosis. Additionally, further research and development of advanced imaging techniques and computational methods could improve the differentiation between these two entities in the future. |

Table 7: Example answers generated by ChatGPT, ChatGPT with our self-reflection loop, and GPT4. We **underline and bold** the concluding parts aligning with the golden answer and *underline and italicize* the concluding parts conflicting with the golden answer.