# OpenReview forum: "Towards Mitigating LLM Hallucination via Self Reflection"
_EMNLP/2023/Conference — EMNLP 2023 Findings_

### Official Review · Reviewer_tvQQ · 2023-08-04

**Soundness:** 4

**Excitement:**

4: Strong: This paper deepens the understanding of some phenomenon or lowers the barriers to an existing research direction.

**Paper Topic And Main Contributions:**

The paper presents a novel knowledge loop design, primarily employed in the prompting process, with a focus on enhancing knowledge consistency during the inference process of large models. The paper has tested the performance across multiple datasets, most of which demonstrate a certain degree of improvement.

**Questions For The Authors:**

See above section.

**Reasons To Accept:**

1. The paper introduces a novel self-enhancement method for knowledge, which enhances its reliability in terms of knowledge;
2. The paper conducts evaluations on multiple datasets, especially those involving scenarios related to the application of large models (such as in medicine), demonstrating the practical value of the model;
3. The paper provides a wealth of reproducible details, lending strong credibility to the study.

**Reasons To Reject:**

The designed method  does not appear to have inherent advantages in terms of time efficiency and performance stability.

In fact, the model's ability to generate different types of knowledge is not consistent (for instance, its performance tends to be poorer on long-tail knowledge). For knowledge areas where the model does not excel, the paper may not have conducted a more in-depth analysis of potential error propagation.

**Reproducibility:**

5: Could easily reproduce the results.

**Reviewer Confidence:**

3: Pretty sure, but there's a chance I missed something. Although I have a good feel for this area in general, I did not carefully check the paper's details, e.g., the math, experimental design, or novelty.

---

> ### Author Rebuttal · Authors · 2023-08-29
>
> ## Response to “Reason 1 To Reject about time efficiency and performance stability”:
> Thanks for your valuable feedback! We carefully design guides/instructions for refinement so that reduce the cycle number and improve the time efficiency. We acknowledge that compared to direct generation, our method reduces hallucinations but sacrifices time efficiency. And sampling during generation introduces some randomness. Our future will explore a better guide for refinement or during inference while maintaining the accuracy of QA systems.
>
> ## Response to “Reason 2 To Reject about different types of knowledge”:
> Thanks for your exciting suggestion! We notice the long-tail problem during the experiment analysis and believe it’s an interesting topic to explore. Due to the limited number of problems addressed by one work, we do not treat it as our main point. Our goal is to enhance generated performance as much as possible within the context of mastering knowledge. In future work, we will improve our current method to treat different types of knowledge/questions differently. We will dig into the deeper causes in knowledge areas. Specifically, we think augmenting with external knowledge or adjusting the distribution of the training dataset can be an effective method.

---

### Official Review · Reviewer_coTz · 2023-08-05

**Soundness:** 3

**Excitement:**

2: Mediocre: This paper makes marginal contributions (vs non-contemporaneous work), so I would rather not see it in the conference.

**Paper Topic And Main Contributions:**

This paper analyzes the hallucination problem of large models in the task of generative medical QA, and proposes a method of generating knowledge, verifying knowledge, and generating answers based on knowledge. The experiment proves that the reflection method proposed in this paper can effectively improve the quality of the answers generated by the existing model.


**Reasons To Accept:**

1. For motivations, this paper analyzes the task of large language models in the task of generative medical QA, and summarizes the categories of hallucinations produced by large models in this scenario. This gives us a guiding significance for designing generative medical QA models.

2. For the design of the method, this paper proposes a method of generating answers based on knowledge by using the language model to generate the knowledge corresponding to the answer to the question first, continuously verifying the confidence of the generated knowledge, and finally using the generated knowledge. This method is cleverly designed, and the experimental results also prove the effectiveness of this method.

3. The article has done sufficient experiments to verify that this method is effective on many modern language models, which also shows the versatility of this method.

**Reasons To Reject:**

1. Regarding the novelty, the method of this paper has been verified by some existing work. For example, the LLM is allowed to generate knowledge first and then answer. You can refer to these two papers (https://arxiv.org/abs/2210.01296, https://arxiv.org/pdf/2209.10063); about Self-Reflection can refer to this paper (https://arxiv.org/abs/2303.11366). The author needs to elaborate on the uniqueness of this paper compared to these methods. Of course, the reviewer agrees with the necessity of building a credible QA model in the medical field.

2. Regarding the soundness of the method, this paper needs to explain the reliability of using the language model as the knowledge source, especially to explain whether using a general language model can provide better and more reliable knowledge than using a medical knowledge base?

**Reproducibility:**

4: Could mostly reproduce the results, but there may be some variation because of sample variance or minor variations in their interpretation of the protocol or method.

**Reviewer Confidence:**

3: Pretty sure, but there's a chance I missed something. Although I have a good feel for this area in general, I did not carefully check the paper's details, e.g., the math, experimental design, or novelty.

**Typos Grammar Style And Presentation Improvements:**

The title is not feasible for this paper.
The authors should consider more narrative titles, such as ``Towards Mitigating Hallucination in Medical Question Answering with Large Language Models via Self-Reflection''

---

> ### Author Rebuttal · Authors · 2023-08-29
>
> Thanks for your valuable feedback!
>
> ## Response to “Reason 1 To Reject about novelty”:
> We introduce an interactive self-reflection methodology that incorporates knowledge acquisition and answer generation. [1-2] propose “generate-then-answer” pipelines without feedback and interactive loop. They do not explore harnessing the inner self-reflection, interactivity, and multitasking ability of LLM.
>
> Similarly, Reflexion[3] explores the self-reflection of LLMs. However, this contemporary work was preprinted on June 10, 2023, and our paper was submitted before June 23. Thus, this work should not undermine our novelty. Moreover, [3] only evaluates and refines the final generated text, while our method considers both the intermediate and the final generated text during generation. In more detail, the feedback in [3] is a label while our feedback for refinement includes Aspect Description and Explicit Score Number. We also do ablation studies for these components. Our method fosters a dynamic interaction between the system and its knowledge, progressively enhancing the model's ability to provide factuality, consistency, and entailment in healthcare settings. We hope our work can contribute to a new pattern to utilize the inner knowledge, interactivity, and self-reflection of LLMs and tap into their potential.
>
> In addition, these arxiv papers are not peer-reviewed.
>
> > [1] Sun, Zhiqing, et al. "Recitation-augmented language models." arXiv preprint arXiv:2210.01296 (2022).
>
> > [2] Yu, Wenhao, et al. "Generate rather than retrieve: Large language models are strong context generators." arXiv preprint arXiv:2209.10063 (2022).
>
> > [3] Shinn, Noah, et al. "Reflexion: Language agents with verbal reinforcement learning." arXiv preprint arXiv:2303.11366 (2023).
>
> ## Response to “Reason 2 To Reject about soundness”:
>
> LMs are trained on a wide variety of data from the internet including medicine-related knowledge. Current works [1-2] explore the potential of LM to become a knowledge source. LLMs achieve impressive performance in knowledge-intensive tasks including the medical domain [3-5].
>
> However, currently, a general LM is still not trustworthy enough as a medical knowledge base in practical applications. That’s the reason why we do this work.
> A manually calibrated external medical knowledge base is more reliable. However, the motivation for our work (retrieval-free) and retrieval-augmented generation (using medical knowledge bases) is different:
> 1. Retrieval-augmented works emphasize that the inherent knowledge in LMs is deficient, and they add external knowledge to compensate.
> 2. Our work aims to harness and maximize the potential of the model's inherent knowledge. Current research works [6-8] highlight a gap between surface realization and inherent knowledge in NLG tasks. Models can realize they are generating something hallucinated/nonsensical in some way. Our method aims to reduce this gap and enhance performance as much as possible within the context of mastering knowledge. As base LLMs become more powerful and inherent knowledge becomes more extensive and versatile, it is crucial to leverage knowledge and unleash its capabilities.
>
> As we said in the Limitation section, our method is still in its early stages and not yet ready for direct real-world deployment. It should be viewed as a complementary approach alongside other methods, such as retrieval, with the potential to contribute to more robust application systems in the future.
>
> > [1] Wang, Cunxiang, Pai Liu, and Yue Zhang. "Can Generative Pre-trained Language Models Serve As Knowledge Bases for Closed-book QA?." Proceedings of the 59th Annual Meeting of the Association for Computational Linguistics and the 11th International Joint Conference on Natural Language Processing (Volume 1: Long Papers). 2021.
>
> > [2] AlKhamissi, Badr, et al. "A review on language models as knowledge bases." arXiv preprint arXiv:2204.06031 (2022).
>
> > [3] Nori, Harsha, et al. "Capabilities of gpt-4 on medical challenge problems." arXiv preprint arXiv:2303.13375 (2023).
>
> > [4] Gilson, Aidan, et al. "How does ChatGPT perform on the United States medical licensing examination? The implications of large language models for medical education and knowledge assessment." JMIR Medical Education 9.1 (2023): e45312.
>
> > [5]  Liévin, Valentin, Christoffer Egeberg Hother, and Ole Winther. "Can large language models reason about medical questions?." arXiv preprint arXiv:2207.08143 (2022).
>
> > [6] Yin, Zhangyue, et al. "Do Large Language Models Know What They Don't Know?." Findings of the Association for Computational Linguistics: ACL 2023.
>
> > [7] Burns, Collin, et al. "Discovering latent knowledge in language models without supervision." International Conference on Learning Representations (ICLR), 2023.
>
> > [8] Rajpurkar, Pranav, Robin Jia, and Percy Liang. "Know What You Don’t Know: Unanswerable Questions for SQuAD." Proceedings of the 56th Annual Meeting of the Association for Computational Linguistics (Volume 2: Short Papers). Association for Computational Linguistics, 2018.
>
> ## Response to “Typos Grammar Style And Presentation Improvements:
> Thanks for your valuable suggestion! We will change the title if it’s allowed in the camera-ready version.

---

### Official Review · Reviewer_8mL5 · 2023-08-08

**Soundness:** 3

**Excitement:**

3: Ambivalent: It has merits (e.g., it reports state-of-the-art results, the idea is nice), but there are key weaknesses (e.g., it describes incremental work), and it can significantly benefit from another round of revision. However, I won't object to accepting it if my co-reviewers champion it.

**Paper Topic And Main Contributions:**

This paper investigates the effect of hallucination in medical question-answering systems. This is indeed an interesting topic, very timely and essential. Mostly this paper is easy to follow and understand. However, I have technical queries - asking relevant questions accordingly.

**Questions For The Authors:**

This is a nice paper overall, but II have the following question.
1. Choices of LLM are not justified well. PaLM etc are not included - not sure why?
2. "After analyzing 250 directly generated examples from the five models" - I argue this is a very small set to conclude anything!
3. I am not sure that I understand this - how MedNLI, CtrlEval, F1 etc. reflect there is no hallucination in the follow-up generation - please clarify.
4. The overall annotation process, inter-annotator agreement, number of annotators etc are completely missing.

**Reasons To Accept:**

With the proliferation of LLM and their merging abilities, the emerging abilities of lying are also increasing. Hallucination could be detrimental in mission-critical applications like medical question-answering. This paper is quite timely and relevant.

**Reasons To Reject:**

I have specific technical queries.

**Reproducibility:**

3: Could reproduce the results with some difficulty. The settings of parameters are underspecified or subjectively determined; the training/evaluation data are not widely available.

**Reviewer Confidence:**

5: Positive that my evaluation is correct. I read the paper very carefully and I am very familiar with related work.

---

> ### Author Rebuttal · Authors · 2023-08-29
>
> ## Response to “Question 1 about the choice of LLM”:
>
> Our setting is zero-shot, and most LLMs (Vicuna, Alpaca, MedAlpaca, Robin) we select are based on LLaMa.  Existing papers/blogs [1-2] compare current LLMs and LLaMa achieves the best zero-shot performance at a similar size. LLaMa and its variants are one of the most commonly used open-source LLMs and achieve impressive performance.
> It's worth noting that, PaLM is more suitable for few-shot learning [3] which does not align with our work.
>
> > [1] Touvron, Hugo, et al. "Llama: Open and efficient foundation language models." arXiv preprint arXiv:2302.13971 (2023).
>
> > [2] https://ai.meta.com/resources/models-and-libraries/llama/
>
> > [3] Chowdhery, Aakanksha, et al. "Palm: Scaling language modeling with pathways." arXiv preprint arXiv:2204.02311 (2022).
>
> ## Response to “Question 2 about the number of the case study (human annotation)”:
>
> The analysis for the case study is extremely time and human source consuming. There are many perspectives to review, and the funding is limited. It takes over one hour to check each example sentence by sentence. The evaluation of 250 QA pairs, comprising roughly 2000 sentences, is conducted by experts and we also did crowd-sourced evaluations as a supplement, costing over 750 person-hour and 1500 USD. Existing works [1-4] also use examples of the same magnitude to analyze (about 100 examples). We are currently asking for more funding to analyze more cases.
>
> > [1] Yang, Shiquan, et al. "An Interpretable Neuro-Symbolic Reasoning Framework for Task-Oriented Dialogue Generation." Proceedings of the 60th Annual Meeting of the Association for Computational Linguistics (Volume 1: Long Papers). 2022.
>
> > [2] Yin, Zhangyue, et al. "Do Large Language Models Know What They Don't Know?." Findings of the Association for Computational Linguistics: ACL 2023.
>
> > [3] Dan Su, et al. Read before generate! faithful long form question answering with machine reading. In Findings of the Association for Computational Linguistics: ACL 2022.
>
> > [4] Bang, Yejin, et al. "A multitask, multilingual, multimodal evaluation of chatgpt on reasoning, hallucination, and interactivity." arXiv preprint arXiv:2302.04023 (2023).
>
> > [5] Liévin, Valentin, Christoffer Egeberg Hother, and Ole Winther. "Can large language models reason about medical questions?." arXiv preprint arXiv:2207.08143 (2022).
>
> ## Response to “Question 3 about the metrics”:
>
> Currently, there is no recognized/common-used metric for hallucination in QA [1]. We try our best to find as many dimensions of metrics as possible to evaluate the hallucination level.
>
> Some works [2-3] in other NLG fields leverage the NLI-based metric to evaluate hallucinations. These metrics define the hallucination score to be the entailment probability between the source/reference and its generated text. These metrics measure the proportion of generated instances that entail, neutral to, and contradict the source/reference. Inspired by them, we introduce MedNLI (Medical Natural Language Inference) [4] to assess the logical entailment of generated answers with the provided context and the reference answer. If the result is “contradiction”, it is identified as “hallucination”.
>
> CTRLEval [5] is an unsupervised, reference-free, and task-agnostic evaluation metric that assesses generation from various aspects (coherence, consistency, and attribute relevance). We focus on the consistency aspect. If the generated answer is factually inconsistent with the context, it is identified as “hallucination”.
>
> Uni-gram F1 and RougeL are classical metrics for Generative QA [6] which reflect the overall performance.
>
> > [1] Lee, Hwanhee, et al. "KPQA: A Metric for Generative Question Answering Using Keyphrase Weights." Proceedings of the 2021 Conference of the North American Chapter of the Association for Computational Linguistics: Human Language Technologies. 2021.
>
> > [2] Laban, Philippe, et al. "SummaC: Re-Visiting NLI-based Models for Inconsistency Detection in Summarization." Transactions of the Association for Computational Linguistics 10 (2022): 163-177.
>
> > [3] Dušek, Ondřej, and Zdeněk Kasner. "Evaluating Semantic Accuracy of Data-to-Text Generation with Natural Language Inference." Proceedings of the 13th International Conference on Natural Language Generation. 2020.
>
> > [4] Phan, Long N., et al. "Scifive: a text-to-text transformer model for biomedical literature." arXiv preprint arXiv:2106.03598 (2021). https://huggingface.co/razent/SciFive-large-Pubmed_PMC-MedNLI
>
> > [5] Ke, Pei, et al. "CTRLEval: An Unsupervised Reference-Free Metric for Evaluating Controlled Text Generation." Proceedings of the 60th Annual Meeting of the Association for Computational Linguistics (Volume 1: Long Papers). 2022.
>
> > [6] Dan Su, et al. Read before generate! faithful long form question answering with machine reading. In Findings of the Association for Computational Linguistics: ACL 2022.
>
>
> ## Response to “Question 4 about the annotation process”:
> We describe the inter-annotator agreement in Section 4.2.2 (Line 497-502) and the details about human evaluation are in Appendix D (Line 1120-1161). Recognizing that you overlooked this part of our paper, we sincerely hope you improve your score!

---

### Official Review · Reviewer_5JJj · 2023-08-09

**Typos Grammar Style And Presentation Improvements:** 1. Section 3.3 title -- possible spel…
**Soundness:** 3

**Excitement:**

4: Strong: This paper deepens the understanding of some phenomenon or lowers the barriers to an existing research direction.

**Paper Topic And Main Contributions:**

This paper studies the existence and mitigation of hallucination in answers generated by LLMs in generative question answering in the medical domain. In particular, the authors first classify the hallucinated answers into 3 types and report their distribution in answers generated by 5 LLMs. Next, an iterative refinement method is proposed that reduces the amount of hallucination by first refining a generated background knowledge and then refining the generated answer based on the background knowledge. Both automatic and human evaluation results are reported and the proposed method seem to reduce the amount of hallucinated answers.

**Questions For The Authors:**

1. How are the thresholds for factuality score and consistency score chosen?

2. Did you consider retrieving related knowledge from a corpus using information retrieval techniques from a reference corpus instead of generating the knowledge with a model?

**Reasons To Accept:**

1. The paper provides a detailed insight and analysis on the existence of hallucination by several state-of-the-art LLMs in generative question answering in the medical domain. The tendency to hallucinate is a critical weakness in the LLMs and this work takes a step towards addressing this.

2. An innovative method is proposed that does not use any external references to mitigate hallucination. The method can prove to be useful for QA in other domains as well.

**Reasons To Reject:**

Original review:

My main concern is with the method for generating knowledge and its refinement. We know that LLMs themselves have a tendency to generate hallucinated text. Thus, it may not be ideal to rely on LLMs to generate knowledge. I understand that the generated knowledge goes through a refinement to ensure factuality. However, the factuality scorer does not compare the generated knowledge to any reference (e.g., looking at an existing knowledge base/corpus). The score is predicted based on a model that is instruction-tuned in a few-shot setting. In other words, the method relies on the model to predict how factual the knowledge is. At this point, the reliability of the model in predicting the factuality score is unclear. A correlation study between the model’s predictions of the factuality score and human evaluation would be useful to verify this.

If the factuality scorer is unreliable, this will impact the answer generation as well since the aim is to make the answer grounded in the final knowledge.

While both automatic and human evaluation shows a decrease in hallucinations in the answers, it is also essential to evaluate the knowledge generation module more carefully.

After rebuttal:

While the rebuttal clarifies and reduces my concerns, I would still like to see a correlation study in this paper for the medical domain. I have however, increased my original soundness score from 2 to 3.

**Reproducibility:**

4: Could mostly reproduce the results, but there may be some variation because of sample variance or minor variations in their interpretation of the protocol or method.

**Reviewer Confidence:**

4: Quite sure. I tried to check the important points carefully. It's unlikely, though conceivable, that I missed something that should affect my ratings.

---

> ### Author Rebuttal · Authors · 2023-08-29
>
> Thanks for your valuable feedback!
>
> ## Response to “Reason To Reject about factuality scorer”:
>
> The motivation for our work (retrieval-free) and retrieval-augmented generation is different:
> 1. Retrieval-augmented works emphasize that the inherent knowledge in LMs is deficient, and they add external knowledge to compensate.
> 2. Our work aims to harness and maximize the potential of the model's inherent knowledge. Current research works [1-5] highlight a gap between surface realization and inherent knowledge in NLG tasks. Models can realize they are generating something hallucinated/nonsensical in some way. Our method aims to reduce this gap and enhance the performance of the model itself. As base LLMs become more powerful and inherent knowledge becomes more extensive and versatile, it is crucial to leverage knowledge and unleash its capabilities.
>
> Given our distinct motivation, we do not use external information during generation, which is similar to chain-of-thought (CoT). Thus, our factuality scorer does not refer to the existing knowledge base. If external knowledge is used for scoring, it is equivalent to introducing knowledge retrieval, which is unfair to the baselines.
>
> Correlation studies between the factuality scorer and human evaluation were conducted by  [6] to verify the effectiveness of in-context instruction learning in diverse NLG tasks. To supplement, we calculate the average factuality score of factual and non-factual cases classified by humans: Factual cases attain -0.89, which is higher than the average score of non-factual (-1.30). This result shows that the factuality score is positively correlated with human evaluation.
>
> On the other hand, our primary emphasis lies on the interactive self-reflection pipeline itself instead of the factuality scorer. Since the scorer is replaceable, especially with emerging and more reliable alternatives that may become available in the future.
>
>
> > [1] Yin, Zhangyue, et al. "Do Large Language Models Know What They Don't Know?." Findings of the Association for Computational Linguistics: ACL 2023.
>
> > [2] Burns, Collin, et al. "Discovering latent knowledge in language models without supervision." International Conference on Learning Representations (ICLR), 2023.
>
> > [3] Rajpurkar, Pranav, Robin Jia, and Percy Liang. "Know What You Don’t Know: Unanswerable Questions for SQuAD." Proceedings of the 56th Annual Meeting of the Association for Computational Linguistics (Volume 2: Short Papers). Association for Computational Linguistics, 2018.
>
> > [4] Kadavath, Saurav, et al. "Language models (mostly) know what they know." arXiv preprint arXiv:2207.05221 (2022).
>
> > [5] Manakul, Potsawee, Adian Liusie, and Mark JF Gales. "Selfcheckgpt: Zero-resource black-box hallucination detection for generative large language models." arXiv preprint arXiv:2303.08896 (2023).
>
> > [6] Fu, Jinlan, et al. "Gptscore: Evaluate as you desire." arXiv preprint arXiv:2302.04166 (2023).
> ## Response to “Question 1 about how to choose threshold”:
>
> The thresholds are hyperparameters, which are empirical values obtained through extensive experiments. First, we sample 250 generated knowledge/answers and calculate their corresponding factuality/consistency scores. Then, we manually assess their quality to classify knowledge into factual or non-factual and classify answers into consistent or inconsistent. This process allows us to find the score gap between the two classes and derive approximate thresholds for the scores. Then, we tune these hyperparameters to optimize the overall performance outcome.
>
> ## Response to “Question 2 about retrieval-related method”:
> Yes. As outlined in the Introduction and Limitation sections, our method is parallel to the retrieval-related methods and it can be combined with other methods. It should be viewed as a complementary approach alongside other techniques including retrieval, contributing to more robust application systems in the future.
>
> As mentioned above, our aim is to explore and maximize the inherent potential of the LLMs, which is different from retrieval-augmented generation. We aim to optimize the performance as much as possible within the context of mastering knowledge.  In cases involving knowledge beyond mastery, such as real-time information, we believe retrieval is an effective way.
>
> ## Response to “Typos Grammar Style And Presentation Improvements:
> Thanks for the correction!

---

### Meta-Review · Area_Chair_JRyy · 2023-09-23

**Recommendation:** 4

**Metareview:**

This paper studies the existence and mitigation of hallucination in answers generated by LLMs in generative question answering in the medical domain. In particular, the authors first classify the hallucinated answers into 3 types and report their distribution in answers generated by 5 LLMs. Next, an iterative refinement method is proposed that reduces the amount of hallucination by first refining a generated background knowledge and then refining the generated answer based on the background knowledge. Both automatic and human evaluation results are reported, and the proposed method seems to reduce the amount of hallucinated answers.

The proposed method has been evaluated on top of Vicuna-7B and ChatGPT, showing improved results on 5 biomedical datasets.

During the discussion period, most questions have been answered reasonably well. One of the reviewers complained about novelty, but the paper's focus on the biomedical domain with its unique challenges makes it different from previous work.

---

### Decision · Program_Chairs · 2023-10-07

**Decision:**

Accept-Findings

**Comment:**

This paper studies the existence and mitigation of hallucination in answers generated by LLMs in generative question answering in the medical domain. In particular, the authors first classify the hallucinated answers into 3 types and report their distribution in answers generated by 5 LLMs. Next, an iterative refinement method is proposed that reduces the amount of hallucination by first refining a generated background knowledge and then refining the generated answer based on the background knowledge. Both automatic and human evaluation results are reported, and the proposed method seems to reduce the amount of hallucinated answers.

The proposed method has been evaluated on top of Vicuna-7B and ChatGPT, showing improved results on 5 biomedical datasets.

During the discussion period, most questions have been answered reasonably well. One of the reviewers complained about novelty, but the paper's focus on the biomedical domain with its unique challenges makes it different from previous work.